# Can Mamba In-Context Learn Task Mixtures?

Yingcong Li [1]  Xupeng Wei [1]  Haonan Zhao [2]  Taigao Ma [2]

## Abstract

In-context learning (ICL) refers to the ability to perform new tasks based on a prompt sequence consisting of "in-context" input-output pairs, without explicit model training. Previous work has shown that State-Space Models (SSMs), particularly Mamba, are potential competitors over Transformers in ICL. However, the capability to handle mixed tasks in complicated ICL prompts remains unanswered. In this work, we explore the Mamba performance in mixed ICL tasks, in a degree from low to high, and from labeled to unlabeled, compared to that of Transformers. We show that Mamba is capable of learning ICL mixtures, reaching the performance of single ICL task and Transformer baselines. Moreover, Mamba converges faster and shows more stable performances than Transformers, allowing Mamba to handle longer context lengths and more complicated prompt structures. Different learning dynamics in different ICL tasks are also observed.

## 1. Introduction

In-context learning (ICL) refers to the ability to formulate predictions through the paired input-output examples that compose the central context of the prompt. Pretrained large language models (LLMs) can efficiently perform ICL from a query input after a few task demonstrations (Brown et al., 2020; Kaplan et al., 2020; Muennighoff et al., 2023). Recently, state-space models (SSMs), particularly Mamba (Gu & Dao, 2023), are potential alternatives to Transformers due to their state-of-the-art performance with a linear time cost. It has been shown that most SSMs are capable of ICL, reaching the performance of the Transformers across various tasks (Akyürek et al., 2024; Park et al., 2024; Grazzi et al., 2024). Although shedding light on whether attention-free LLMs can perform ICL, the previous work has not shown if SSMs can handle the complexity brought by mixtures of task families, which can reflect models' performance on real-world applications where the prompts are composed of a wide range of context-based task mixtures. Specifically, we ask:

> *Can Mamba in-context learn task mixtures? How is Mamba compared to the Transformers in task mixture ICL? What are the learning dynamics in solving task mixture ICL problems?*

Answering these questions helps evaluate the performance of SSMs on future real-world applications where the capability of handling complicated downstream task structures is required. In this work, we pretrain Mamba and Transformer models on the task mixtures, each of which is a function class such as Noisy Linear Regression, ReLU Regression, and Binary Classification. Three different methods of task mixing are compared, including vanilla task mixtures (sequence-level), concatenated task mixtures (block-level), and blended task mixtures (position-level). We also compare three different labeling methods, including no labeling, prompt labeling, and embedding labeling. Our results have shown that:

- Mamba has achieved competitive results close to that of the baseline of single ICL task and Transformer in all different types of task mixtures.

- With either prompt or embedding labeling, both Mamba and Transformer models achieve test risks or accuracy scores comparable to single-task baselines, better than unlabeled plain prompt sequences. However, Transformers fail to generalize to longer context lengths, showing Mamba can better handle unseen longer contexts during test than Transformers.

- Mamba models converge faster than Transformers during training and have no delayed learning behaviors that are observed in Transformer training. Moreover, the convergence rates of solving different task families using Transformer models follow: Noisy Linear Regression > ReLU Regression > Binary Classification, implying a hierarchy of learning difficulty for these tasks in task mixture ICL problems.

---

[1]Department of EECS, University of Michigan, Ann Arbor, MI, USA [2]Department of Physics, University of Michigan, Ann Arbor, MI, USA. Correspondence to: Yingcong Li <yingcong@umich.edu>.

*Proceedings of the 1st Workshop on In-Context Learning at the 41st International Conference on Machine Learning*, Vienna, Austria. 2024. Copyright 2024 by the author(s).

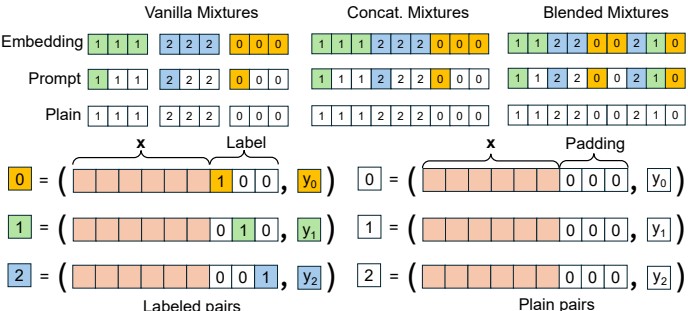

Figure 1: Task mixing problems and labeling methods. Vanilla mixtures are a distribution of single-task prompt sequences in the training and test datasets. Concatenated mixtures fuse sequence blocks of different tasks in the same prompt sequence. Blended mixtures have totally random input-output pairs in each position of the prompt sequence. For label in embedding, all input-output pairs are labeled with the task number. For label in prompt, only the first input-output pair in the same task block is labeled. For plain sequence, no inpu-output pairs are labeled. We use one-hot label for the input-output pairs. Unlabeled input-output pairs are padded with zeros.

## 2. Problem Statement

**In-Context Learning.** In-context learning (ICL) refers to the ability of a model to learn from the prompt sequence consisting of in-context examples (input-output pairs from some tasks) and generate output based on a new query input without any explicit retraining or finetuning. Formally speaking, given a prompt with input-output pairs in the following manner:

$$\boldsymbol{p}_n(f) = (\boldsymbol{x}_i, \boldsymbol{y}_i)_{i=1}^n \quad \text{where} \quad \boldsymbol{y}_i = f(\boldsymbol{x}_i). \quad (2.1)$$

Here, $f \in \mathcal{F} : \mathcal{X} \to \mathcal{Y}$ is the function/task to be learned, and it remains constant in a single prompt but can vary across multiple prompts; $n$ specifies the number of ICL examples in a prompt. The goal of ICL is that for a give test query input $\boldsymbol{x}_q$, the model output would satisfy $\text{Model}(\boldsymbol{p}_n(f), \boldsymbol{x}_q) \approx f(\boldsymbol{x}_q)$.

**Task mixture problems.** Tripuraneni et al. 2023 divides the dataset into a mixture of different tasks, and the prompt sequence contains only one function family with one parameter distribution. Here, we focus on the learning of different types of tasks and fix the task distribution. Specifically, we consider the following types of mixtures depicted in Fig. 1:

- Vanilla task mixture: Each prompt sequence contains $(\boldsymbol{x}_i, \boldsymbol{y}_i)$ pairs generated using the same function $f$, $y_i = f(\boldsymbol{x}_i)$, and a test query, $\boldsymbol{x}_{\text{test}}$ is attached to the sequence at last. Different prompt sequences are assigned with different task functions $f$ where $f$ can be either noisy linear regression, ReLU regression or binary classification function.

- Concatenated task mixture: Each prompt sequence containing $(\boldsymbol{x}_i, \boldsymbol{y}_i)$ pairs is divided into $\leq 6$ concatenated blocks with random block sizes. A random task type, noisy linear regression, ReLU, or binary classification, is assigned to each block.

- Blended task mixture: Each prompt sequence contains $(\boldsymbol{x}_i, y_i)$ pairs where $y_i$ is randomly sampled from $\{f_1(\boldsymbol{x}_i), f_2(\boldsymbol{x}_i), f_3(\boldsymbol{x}_i)\}$ and $f_1, f_2, f_3$ are the noisy linear, ReLU and binary classification functions.

**Labeling methods.** To study how models behave with or without specifying the task identification, we apply one-hot labels for each task appended to the $\boldsymbol{x}$ (unlabeled $\boldsymbol{x}$ has aligned padding). We compare two labeling methods to the plain, unlabeled sequences, respectively:

- Prompt labeling: Only label the first $(\boldsymbol{x}_i, y_i)$ pair after a task transition.

- Embedding labeling: All $(\boldsymbol{x}_i, y_i)$ pairs are labeled.

## 3. Evaluations

**Experimental setting.** All our empirical results are trained with two different types of models: small GPT-2 and Mamba. The small GPT-2 has 6 layers, 4 attention heads per layer and 128 dimensional embeddings, and Mamba contains 6 layers and 128 dimensional embeddings. We consider three different types of tasks: noisy linear regression, ReLU, and binary classification. More specifically, the in-context sample pairs of $(\boldsymbol{x}, y)$ are generated via:

- Noisy linear regression: $y = \boldsymbol{w}^\top \boldsymbol{x} + z$;

- ReLU: $y = (\boldsymbol{w}^\top \boldsymbol{x})_+$;

- Binary classification: $y = \text{sgn}(\boldsymbol{w}^\top \boldsymbol{x})$.

Here, inputs $\boldsymbol{x}$ and weight parameters $\boldsymbol{w}$ are both $d$-dimensional and randomly sampled from the normal distribution, i.e., $\boldsymbol{x}, \boldsymbol{w} \in \mathbb{R}^d$ and $\boldsymbol{x}, \boldsymbol{w} \sim \mathcal{N}(0, \boldsymbol{I}_d)$ and $z \in \mathbb{R} \sim \mathcal{N}(0, \sigma^2)$ is the random noise. We set $d = 5$ and $\sigma^2 = 0.1$ in our experiments. Let $\hat{y}$ be the model prediction. During training, the squared loss of $(y - \hat{y})^2$ is

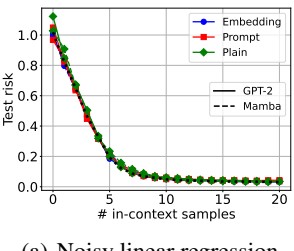
(a) Noisy linear regression

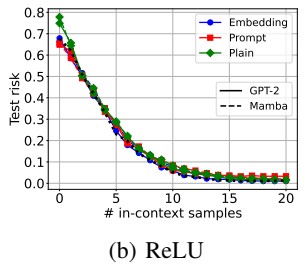
(b) ReLU

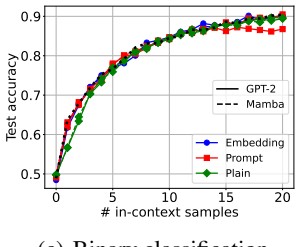
(c) Binary classification

Figure 2: Test risk and accuracy of the Mamba and GPT-2 models on vanilla ICL mixture tasks (solid: GPT-2, dashed: Mamba, black dotted: single-task performance of Mamba model as baselines).

used to train noisy linear regression task and ReLU task, and sigmoid loss is used to train binary classification task. During testing, we report the squared loss as the test risk for both noisy linear regression task and ReLU task, and classification accuracy is reported as the test accuracy for binary classification task. We consider meta learning setting where training samples are randomly generated. For single-task and vanilla mixture ICL problem, each model is trained with 50k iterations with batch size $64$; while to fully train the model to solve the concatenated and blended ICL mixture problems, models are trained with 100k iterations with the same batch size. We use learning rate $10^{-4}$ in all our experiments and normalized risks are reported.

**Single task ICL baselines.** We find Mamba and GPT-2 models share similar performances in all three single-task ICL, see Appendix A. We also find different learning dynamics in three tasks, with the ICL convergence rates following: Noisy linear regression > ReLU regression > Binary classification, indicating the relative difficulties in learning these tasks.

### 3.1. Vanilla ICL Mixture

In Figures 2(a)-2(c), we evaluate and compare Mamba's capability with the vanilla ICL mixture. Three key comparisons are made: Mamba's (dashed) versus GPT-2's (solid) in vanilla ICL mixtures; each model's performance on ICL mixtures versus single-task ICL (black); and different labeling methods - embedding (blue), prompt (red), and plain (green). We observe that Mamba's performance approaches that of GPT-2 in vanilla ICL mixtures, with both models matching the single-task baseline (Appendix A) across all labeling methods. This demonstrates the capability of both models in effectively managing vanilla ICL mixtures. We further examine the benefit of labeling with small number of in-context examples. See Appendix B for details.

### 3.2. Concatenated ICL Mixture

In Figures 3(a)-3(c), we compare the performance of Mamba and GPT-2 models in concatenated ICL mixtures. Similar to the observations in the vanilla case, Mamba's

performance aligns well with that of GPT-2 across all three tasks and three different labeling methods. The performance with labeling, either through embedding or prompt, reach the same level as the single-task baselines. Again, both models without labeling can only show some limited learning abilities compared to the baseline.

To further understand the learning dynamics of ICL mixture, in Figs. 3(d)-3(f), we show the test performance for three tasks individually during training iterations. There are three observations. First, we can see that for both models, plain labeling has a slower converge rate compared to embedding or prompt, meaning that adding labels enhances the ability of both models to more effectively learn the ICL mixtures. Second, different types of tasks have different convergence rates. Noisy linear regression and ReLU tasks converge within 20k and 30k iterations, while binary classification takes a much longer time to converge, i.e., around 40k iterations, suggesting a sequential learning order with noisy linear regression goes first, then ReLU, and finally binary classification. In addition, compared to GPT-2, Mamba consistently converges faster across all tasks. Finally, we notice that during training, GPT-2's test performance also shows a peculiar behavior: it first reaches a plateau, and then exhibits sudden improvements. This highlights different learning styles: Mamba learns in a continuous and gradual manner, whereas GPT-2 demonstrates emergent performance improvements during training.

### 3.3. Blended ICL Mixture

To explore the limit of Mamba's ability in learning ICL mixtures, we look at the performance of the Mamba and GPT-2 in a more challenging setting: blended ICL mixtures, as shown in Figures 4(a)-4(c). We find that adding labels, either through embedding or prompt, can significantly improve the test performance. The notable performance differences between embedding and prompt labeling methods highlight the importance of explicit labeling for each input-output pair in learning task mixtures. We find Mamba closely matches GPT-2 in noisy linear regression, surpasses GPT-2 in ReLU across all labeling methods, and achieves comparable accuracy with GPT-2 under the embedding la-

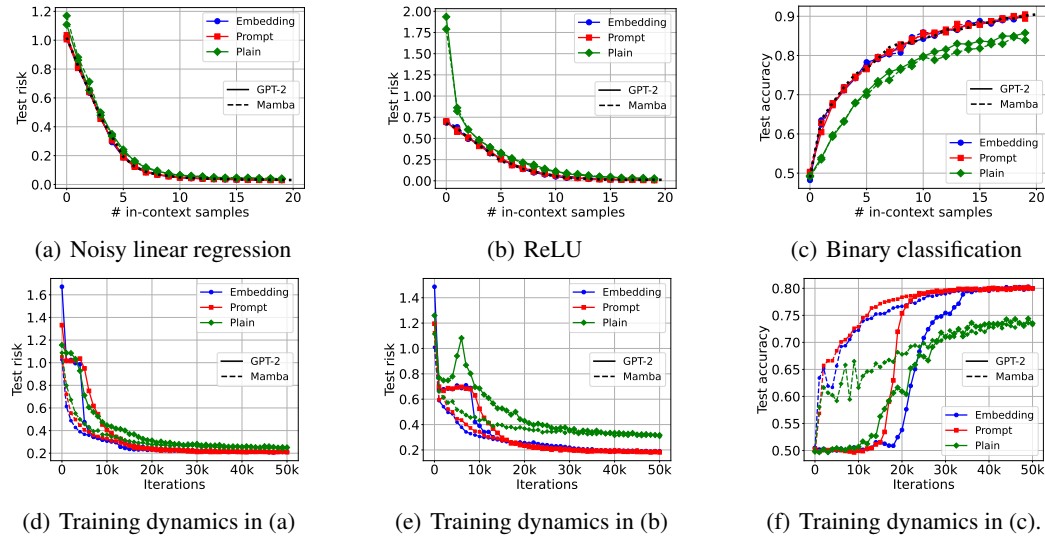

(a) Noisy linear regression      (b) ReLU      (c) Binary classification

(d) Training dynamics in (a)      (e) Training dynamics in (b)      (f) Training dynamics in (c).

Figure 3: The first column shows test risk and accuracy of the Mamba and GPT-2 models on concatenated ICL mixture tasks, and the second column presents the dynamics of performance metrics during training (solid: GPT-2, dashed: Mamba, black dotted: single-task performance of Mamba model).

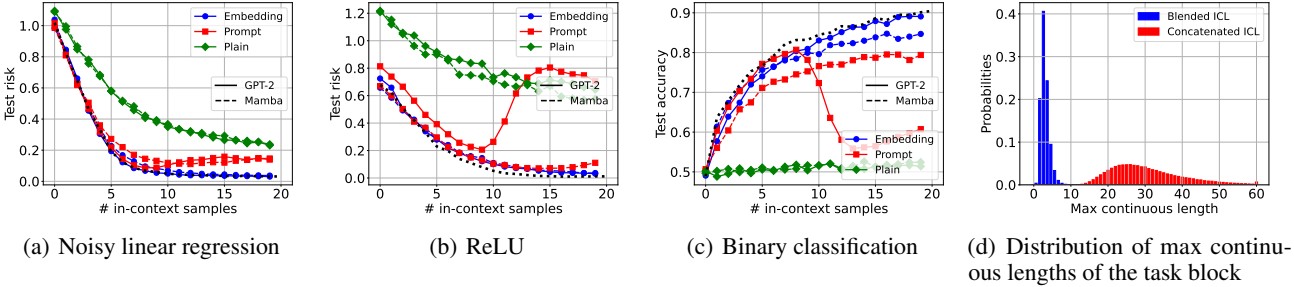

(a) Noisy linear regression      (b) ReLU      (c) Binary classification      (d) Distribution of max continuous lengths of the task block

Figure 4: Test risk and accuracy of the Mamba and GPT-2 models on blended ICL mixture tasks (solid: GPT-2, dashed: Mamba, black dotted: single-task performance of Mamba model).

beling. This suggests Mamba's continued effectiveness in ICL within blended mixtures. Notably, although GPT-2's performance matches or exceeds that of Mamba in all tasks with prompt labeling for a small number of in-context samples, its performance declines when the sample number exceeds 8, unlike Mamba, whose performance continues to improve. This reflects Mamba's superior generalization capabilities compared to GPT-2.

To further verify that the cause of the poor performance of GPT-2 is its inability in extrapolation, we examine the distribution of the maximum length of consecutive tasks of the same type in training sequences, illustrated in Figure 4(d). For sequences comprising 60 input-output pairs, most continuous lengths of the same task are fewer than 8 in blended mixtures, suggesting that the models are unaccustomed to long, uninterrupted blocks of the same task type. This supports the observation that GPT-2 with prompt labeling quickly deteriorates when task block lengths exceed the typical training scenarios.

## 4. Conclusion

In this work, we empirically explore Mamba's ability to learn in-context mixtures with noisy linear regression, ReLU, and binary classification tasks, and compare its performance in three types of ICL mixtures: vanilla, concatenated, and blended ICL mixtures, with three methods: no-label, embedding and prompt. Our results indicate that Mamba achieves a level of performance in learning task mixtures comparable to that of GPT-2, and similarly consistent with single-task baselines. We also notice that different tasks in the ICL mixtures can have different convergence rates, this makes sense as different tasks can have different learning difficulty levels. Notably, Mamba demonstrates faster and more stable convergence than GPT-2, making it potentially more advantageous for real-world applications that require processing diverse task mixtures, which could reduce the computational demands for training and inference. Future work may explore the scalability of Mamba's rapid convergence across extensive ICL task mixtures and the potential benefit of adopting hybrid architectures.

## Acknowledgement

The authors thank Prof. Samet Oymak for helpful guidance and discussions.

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

## A. Baseline: Single-Task ICL

In Figure A1, we report the performance of the Mamba and GPT-2 models on single-task ICL. At first glance, we can see the curves of Mamba and GPT-2 model overlap. This suggests that Mamba is capable of in-context learning different types of tasks with performance close to GPT-2. As the number of in-context samples increases, the test risks for both noisy linear regression and ReLU tasks decreases and approaches zero. Similarly, the test accuracy for the binary classification task increases from $\sim 0.5$ (random guess) to $0.9$. In addition, different types of tasks exhibit different convergence behaviors. The test risks for noisy linear regression and ReLU tasks approach $0$ when the number of in-context samples achieves $10$ and $15$, respectively. However, the test accuracy for binary classification task seems not to converge to $1$ within $20$ in-context examples, suggesting different learning difficulty for the model to learn. This aids in the explanation of the learning dynamics of ICL mixtures.

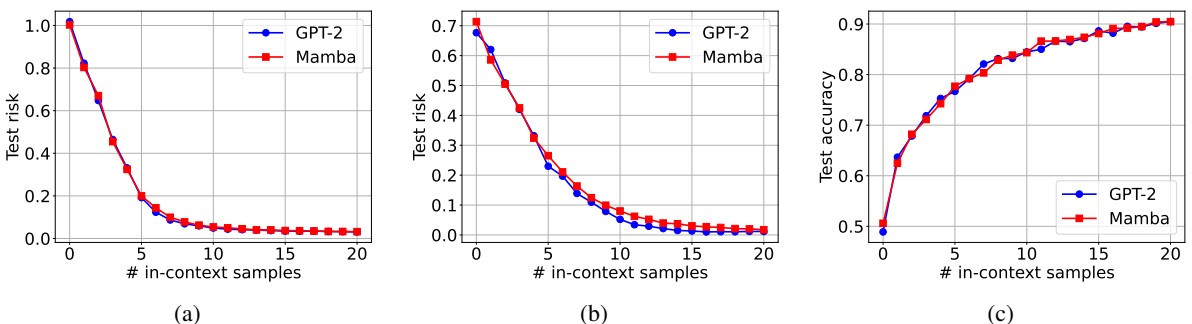

Figure A1: Test risk and test accuracy of the Mamba and GPT-2 models on single-task ICL (baseline). (a) Noisy linear regression. (b) ReLU. (c) Binary classification.

## B. The zoomed-in comparisons in vanilla mixtures

To closely examine the benefits of labeling with a lack of in-context examples, we investigate the test results with small number of in-context samples, i.e., from 0 to 5, as shown in Figures B1. Without labels (plain), both models underperform compared to single-task ICL, most notably with just 1 or 2 in-context examples. Introducing labels via embedding or prompt enhance performance substantially, and align with the baseline performance of single-task ICL. This improvement is logical given that labels provide the model with more context, which aids in task identification.

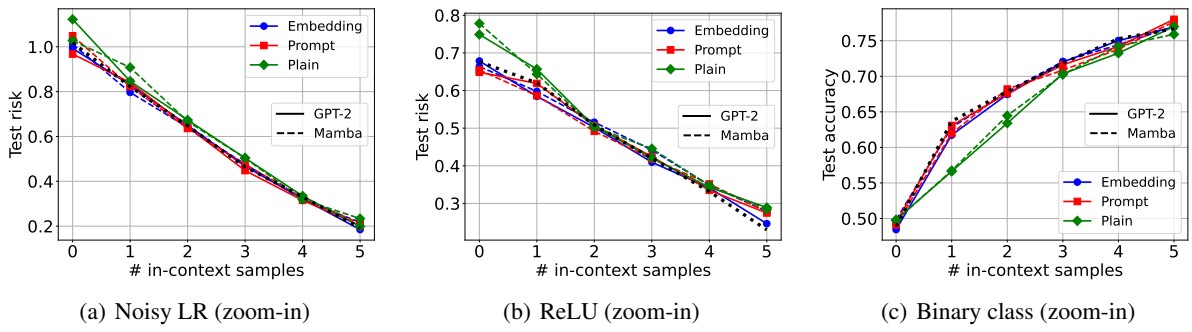

(a) Noisy LR (zoom-in)  (b) ReLU (zoom-in)  (c) Binary class (zoom-in)

Figure B1: Zoom-in of the first few ICL samples of the Mamba and GPT-2 models on vanilla ICL mixture tasks (solid: GPT-2, dashed: Mamba, black dotted: single-task performance of Mamba model as baselines).

