# OpenReview forum: "Can Mamba In-Context Learn Task Mixtures?"
_ICML.cc/2024/Workshop/ICL — ICML 2024 Workshop ICL Poster_

### Official Review · Reviewer_c8GS · 2024-05-31
**Very good paper with great insights**

**Rating:** 3
**Fit:** 3
**Confidence:** 3

**Workshop Review:**

**Summary**

The paper studies and compares Mamba’s performance on task mixtures to transformers in a well-designed empirical study. In particular, they provide initial answers for whether Mamba can do in-context learn on task mixtures, how it compares to transformers, and showcases the different learning dynamics of Mamba compared to transformers.

**Strengths**

* The paper is very well-written and clear (just minor suggestions below). I want to thank the authors for their outstanding effort!

* The paper aligns well with intuition and their findings are important contributions to our understanding of Mamba (and transformers).

* Experiments are well-designed and carefully conducted. They support well the findings of the present paper.

* It’s interesting to see how task complexity influences the learning behavior. Similar experiments for the other task mixture problems would be appreciated.

* The analysis on the generalization problems of GPT-2 is a great insight.

**Weaknesses & questions**

* [W] While all experiments were conducted with d=5, it would be interesting to see what happens for larger d (let’s say up to 20 similar to Garg et al [1]).

* [W] The paper doesn’t hypothesize what leads Mamba to converge faster than GPT-2. Similarly, a discussion on the learning order of tasks would be interesting.

* [Q] The learning behavior of GPT-2 is quite intriguing and makes me wonder if it is connected to gradient flow problems caused by Softmax, as observed by prior work [2,3]? If true, it may also explain Mamba’s faster, more smooth convergence, as Mamba doesn’t have any Softmax. As mentioned above, it would be very interesting to dive deeper into the reasons for the different convergence behavior of Mamba and transformers.

**Suggestions**

* l. 43, left column: I’d rather list the linear time cost as the main benefit of SSMs instead of compact model size (not sure what is meant by that).

* Fig. 1: I’d flip the mixtures & label subfigures to better match the caption.

* l. 91, left column: The citation should be without parentheses as it is part of the text.

* Fig. 1 is not referenced from the main text.

* It‘d be good to show the test error instead of the test accuracy to better align with the errors from the regression tasks.

* Fig. A1: it would be interesting to see the behavior for #in-context examples >> d*4; especially as the binary classification task doesn’t seem to converge.

---

[1] Garg, Shivam, et al. "What can transformers learn in-context? a case study of simple function classes." NeurIPS 2022.

[2] Zhai, Shuangfei, et al. "Stabilizing transformer training by preventing attention entropy collapse." ICML 2023.

[3] Hoffmann, David T., et al. "Eureka-Moments in Transformers: Multi-Step Tasks Reveal Softmax Induced Optimization Problems." arXiv 2023.

**Reason For Not Giving Higher Score:**

N/A

**Reason For Not Giving Lower Score:**

The present paper is well-written with great insights on the behavior of Mamba compared to transformers on mixture tasks. Beyond that, the work outlines an intriguing difference in the learning behavior of Mamba vs. transformers. While the paper could have conducted more thorough analysis, I believe this is okay given the scope of a workshop paper; though I’d appreciate it if the authors could include such in the final and/or full-lengh paper version. Overall, the paper is an important contribution to understanding similarities and differences of transformer and state-space models, which is a very relevant research direction.

---

### Official Review · Reviewer_mGLW · 2024-06-06
**An interesting empirical paper which should include more details.**

**Rating:** 3
**Fit:** 3
**Confidence:** 2

**Workshop Review:**

This paper investigates the ICL of Mamba and Transformers by considering a specific task mixture of noisy linear regressions, ReLU functions and binary classifications. The input sequences takes 3 formats: vanilla task mixture, concatenated task mixture and blended task mixture. Two labeling methods to indicate task (transitions) are also compared.

* Interest to the community: I found the setting of prompt labeling method on blended task mixture particularly interesting. In this setting, the trained Transformer has issue on length generalization while Mamba performs well on long test sequences. There might be some deeper explainations and I hope the authors keep exploring it.

* Clarity: Overall this paper is clearly written though some details are omitted, e.g., whether there is positional embedding in the input sequences. The setting of no labeling in blended task mixture is a bit strange since there should not be any reasonable solution in that case.

* Correctness: I didn't try to replicate the experimental results so I don't know.

**Reason For Not Giving Higher Score:**

3 is the highest score.

**Reason For Not Giving Lower Score:**

The setting of prompt labeling method on blended task mixture is interesting. In this setting, the trained Transformer has issue on length generalization while Mamba performs well on long test sequences. There could be some research following this line to see what is the mechanism behind it.

---

### Meta-Review · Area_Chair_KLS9 · 2024-06-14

**Recommendation:** 2

**Metareview:**

This paper follows up on prior research comparing Mamba to Transformer models on simple machine learning problems, but asks the question whether it can also adapt to task mixtures and not just single tasks.
Both reviewers point out the good writing of the paper and are overwhelmingly positive of the paper.

I recommend acceptance.

---

### Decision · Program_Chairs · 2024-06-17

Accept (Poster)